# Accurate Calibration of a Large Field of View Camera with Coplanar Constraint for Large-Scale Specular Three-Dimensional Profile Measurement

**DOI:** 10.3390/s23073464

**Published:** 2023-03-25

**Authors:** Rongsheng Lu, Zhizhuo Wang, Zhiting Zou

**Affiliations:** 1School of Instrument Science and Opto-Electronics Engineering, Hefei University of Technology, Hefei 230009, China; 2Anhui Province Key Laboratory of Measuring Theory and Precision Instrument, Hefei University of Technology, Hefei 230009, China

**Keywords:** coplanar constraint, camera calibration, large FOV, specular measurement

## Abstract

In the vision-based inspection of specular or shiny surfaces, we often compute the camera pose with respect to a reference plane by analyzing images of calibration grids, reflected in such a surface. To obtain high precision in camera calibration, the calibration target should be large enough to cover the whole field of view (FOV). For a camera with a large FOV, using a small target can only obtain a locally optimal solution. However, using a large target causes many difficulties in making, carrying, and employing the large target. To solve this problem, an improved calibration method based on coplanar constraint is proposed for a camera with a large FOV. Firstly, with an auxiliary plane mirror provided, the positions of the calibration grid and the tilt angles of the plane mirror are changed several times to capture several mirrored calibration images. Secondly, the initial parameters of the camera are calculated based on each group of mirrored calibration images. Finally, adding with the coplanar constraint between each group of calibration grid, the external parameters between the camera and the reference plane are optimized via the Levenberg-Marquardt algorithm (LM). The experimental results show that the proposed camera calibration method has good robustness and accuracy.

## 1. Introduction

In recent years, the vision measurement system has been widely used in industrial production due to its high precision, non-contact, real-time capabilities, etc. [1,2]. At the same time, for some special objects, such as car windshield [3], painted body shell [4], polishing mold, stainless steel products, and other smooth surface objects, the demand for three-dimensional measurement is greater and greater. Meanwhile, the traditional three-dimensional reconstruction method [5,6,7] is not ideal for the reconstruction of the bright surface. The two-dimensional feature information of the image obtained by the camera mainly comes from the surrounding environment of the shiny surface, rather than the surface itself. For the high reflection characteristics of the shiny surface, the reference pattern is usually placed around it, and the reference pattern modulated by the surface helps realize three-dimensional reconstruction of itself [8,9,10,11,12]. In this case, the calibration accuracy of the reference plane and camera directly affects the subsequent three-dimensional reconstruction accuracy of the shiny surface. Meanwhile, to measure more area of surface, a camera with a large FOV is needed. However, for calibration in large FOV, the targets with large areas and high precision are not only difficult to make, but they are also inconvenient to carry and use.

For the calibration of the catadioptric system, many scholars have proposed methods of using an auxiliary plane mirror to estimate the external parameters between the camera and the reference object [13]. Kumar et al. [14] proposed using the orthogonal constraint between the direction vector of the connection from the corresponding point of the object to the mirror image, as well as the column vector of the rotation matrix to list linear equations for solving, and each set of equations requires at least five calibration images. However, the calculated position parameter has a large error with the true value, which is harmful to the subsequent parameter optimization. Takahashi et al. [15] obtain the unique solution of three P3P problems (perspective-three-point problem) from three mirror images based on the orthogonal constraint. However, if the reference object is smaller than a certain size, a wrong solution will be obtained. The method proposed by Hesch et al. [16] also obtains the solutions of three P3P problems from three mirror images, but it can only select an optimal solution from 64 candidate solutions after re-projection error evaluation. Xin et al. [17] directly estimate the camera rotation matrix by the SVD decomposition of the sum of the rotation matrices. Additionally, they calculate the translation vector by solving overdetermined linear equations. While it is more sensitive to noise, the algorithm stability is poor. Bergamasco [18] proposed a method to locate coplanar circles from images by means of a non-cooperative evolutionary game and refined the estimation of camera parameters by observing a set of coplanar circles. However, the accuracy of this method is low.

For the calibration of the camera with a large FOV, scholars consider combing several two-dimensional small targets into a large three-dimensional target. While in the methods proposed in the paper [19,20], all intrinsic parameters of the camera cannot be obtained because the polynomial projection mode is used. Meanwhile, in the methods proposed in the paper [21,22], the relative positions between the small targets are subject to certain restrictions, which makes it difficult to be applied in real applications. Occlusion-resistant markers, such as Charuco [23] or RUNETag [24], are also robust options, but they present fewer points for calibration.

To solve this problem, we use a LCD monitor as a reference plane to produce the calibration grid. It not only solves the problem of difficulty in manufacturing, carrying, and using large-sized objects, but it also can be used as a carrier for projecting encoded patterns when measuring bright surfaces due to its ability to produce free patterns. Bergamasco [25,26] also used a monitor that displays dense calibration grids for camera calibration, but it requires multiple frames, and when dense grid points are spread over the display, the curvature of the display surface will greatly affect the accuracy and robustness of calibration. Therefore, this article calibrates using a smaller calibration grid on the monitor and covers the camera’s field of view by moving the position of the calibration grid, which to some extent reduces the impact of display surface curvature, and ultimately it achieves high accuracy and robustness.

Firstly, by moving the calibration grid on the reference plane and changing the tilt angle of the plane mirror on the optical platform to obtain multiple sets of mirrored calibration images, the internal and external parameters of the camera are computed by Zhang’s [27] calibration method. Secondly, the orthogonality constraint calibration method and P3P algorithm proposed in [15,16] are used to obtain the external parameters from the reference plane to the camera. Finally, the LM [28] algorithm is used to obtain the optimal solution of the external parameters with the coplanar constraint of multiple calibration grid positions. At the same time, using the method of reconstructing the smooth mirror shape from a single image proposed in [12], three-dimensional measurement experiments are carried out to indirectly verify the accuracy of the calibration method proposed.

## 2. Geometry of Camera Pose Estimation

### 2.1. Plane Mirror Reflection Model

As is shown in Figure 1, in the camera coordinate system ***C***, the plane mirror can be described by the plane parameters Π={n,d}. The unit vector n denotes the normal vector of the mirror plane, d represents the distance between the origin of C and the plane [17], and $R_{s2c}$ and $T_{s2c}$ are the rotation matrix and the translation vector between the reference plane coordinate system and the camera coordinate system. P is a feature point on the reference plane.

Based on the reflection property of the mirror, the relationship between this point and its mirror point is given by:(1)$$\left[\begin{array}{c} P^{\prime }\\ 1 \end{array}\right]=M_{1}\cdot \left[\begin{array}{c} P\\ 1 \end{array}\right],M_{1}=\left[\begin{array}{cc} I-2\cdot n\cdot n^{T} & 2\cdot d\cdot n\\ O & 1 \end{array}\right]$$
This denotes the symmetric transformation induced by Π. Note that $M_{1}={M_{1}}^{-1}$, and $\left(I-2\cdot n\cdot n^{T}\right)$ is a Householder matrix. Let $M_{2}$ describe the rigid transformation that transforms points from the reference to the camera frame:(2)$$M_{2}=\left[\begin{array}{cc} R_{s2c} & T_{s2c}\\ O & 1 \end{array}\right]$$

### 2.2. Mirror-Based Camera Projection Model

The perspective projection model is a camera imaging model widely used in computer vision [23]. The mapping relation between any three-dimensional point $P_{w}$ in the space and its corresponding pixel point $v={\left[x\;y\;1\right]}^{T}$ in the image can be described as:(3)$$v=s\cdot A\cdot [R\;T]\cdot P_{w}$$
where s is a nonzero scale factor, A is the intrinsic parameters matrix of the camera, and R and T are the rotation matrix and the translation vector between the camera coordinate system and the world coordinate system. Taking the mirror reflection into account, concatenate the camera model with the mirror reflection, the mirror-based camera projection model becomes: (4)$$v=s\cdot A\cdot M_{1}\cdot M_{2}\cdot P_{w}$$

R and T can be written as:(5)$$\{\begin{array}{l} R=(I-2\cdot n\cdot n^{T})\cdot R_{s2c}\\ T=(I-2\cdot n\cdot n^{T})\cdot T_{s2c}+2\cdot d\cdot n \end{array}$$

According to the Equation (5), we need at least three specular reflection images to calculate $R_{s2c}$ and $T_{s2c}$.

### 2.3. Computation of External Parameters

By changing the tilt angle of the plane mirror, we can obtain mirrored images at different positions and compute external parameters by the P3P algorithm [16]. Let $j,j^{\prime }\in \{1,2,3\}$, and $R_{j}$ represents the rotation matrix of the mirrored image at the j position of the plane mirror. Assume unit vector $m_{jj^{\prime }}$ is perpendicular to $n_{j}$ and $n_{j^{\prime }}$, so we can obtain:(6)$$R_{j}\cdot {R_{j^{\prime }}}^{T}\cdot m_{jj^{\prime }}=(I-2\cdot n_{j}\cdot {n_{j}}^{T})\times (I-2\cdot n_{j^{\prime }}\cdot {n_{j^{\prime }}}^{T})\cdot m_{jj^{\prime }}=m_{jj^{\prime }}$$

$R_{j}\cdot {R_{j^{\prime }}}^{T}$ is a special orthogonal matrix, which has two complex conjugate eigenvalues, and one eigenvalue equals 1. So $m_{jj^{\prime }}$ is the eigenvector of $R_{j}\cdot {R_{j^{\prime }}}^{T}$ corresponding to the eigenvalue of 1. According to the cross-product properties of the eigenvector, the unit normal vectors corresponding to the three positions of the plane mirror can be calculated.
(7)$$n_{1}=\frac{m_{13}\times m_{12}}{\left\|m_{13}\times m_{12}\right\|},\;n_{2}=\frac{m_{21}\times m_{23}}{\left\|m_{21}\times m_{23}\right\|},\;n_{3}=\frac{m_{13}\times m_{23}}{\left\|m_{13}\times m_{23}\right\|}.$$

According to the Equation (5), $R_{s2c}$ can be calculated. In the case of an ideal condition without noise, the three rotation matrices calculated by three $R_{j}$ should be equal. While they are not equal in fact due to the noise. Therefore, the average of the rotation matrices should be calculated [20].
(8)$$\bar{R}={\left[{\left({\hat{R}}^{T}\cdot \hat{R}\right)}^{1/2}\right]}^{-1}\cdot \hat{R},\;\mathrm{where}\;\hat{R}=\frac{1}{3}\cdot \sum_{j=1}^{3}R_{s2cj}$$

The rest of the parameters ${\left[T,\;d_{1},\;d_{2},\;d_{3}\right]}^{T}$ can be solved by linear equations constructed by the Equation (5). So far, all of the initial values of the pose parameters have been calculated.
(9)$$\left[\begin{array}{cccc} \left(I-2\cdot n_{1}\cdot {n_{1}}^{T}\right) & 2\cdot n_{1} & o & o\\ \left(I-2\cdot n_{2}\cdot {n_{2}}^{T}\right) & o & 2\cdot n_{2} & o\\ \left(I-2\cdot n_{3}\cdot {n_{3}}^{T}\right) & o & o & 2\cdot n_{3} \end{array}\right]\cdot \left[\begin{array}{c} T\\ d_{1}\\ d_{2}\\ d_{3} \end{array}\right]=\left[\begin{array}{c} T_{1}\\ T_{2}\\ T_{3} \end{array}\right]$$

### 2.4. Optimization with Coplanar Constraint

Linear solutions are usually sensitive to noise, we can minimize the reprojection error of back-projection by adjusting $R_{s2c}$, $T_{s2c}$, n and d with coplanar constraint. As is shown in Figure 2, we move the calibration grid on the LCD monitor for W times and rotate the plane mirror corresponding to each grid position for M times. The grid has N characteristic corners. Let $R_{\mathrm{ji}}$ represent the rotation matrix of the mirrored image of the j grid at the i plane mirror position. In the same way, $T_{\mathrm{ji}}$ is translation vector, $n_{\mathrm{ji}}$ represents the normal vector of the mirror, $d_{ji}$ represents the distance between the origin of the camera coordinate system and the plane mirror, $R_{s2cj}$ represents rotation matrix from the j checkerboard coordinate system to the camera coordinate system, and $T_{s2cj}$ represents the translation vector. $P_{k}$ represents the k feature point of the grid in the reference plane coordinate system. $q_{\mathrm{jik}}$ represents the projection point of the k feature point of the j grid at the i planar mirror position. ${\tilde{q}}_{\mathrm{jik}}$ represents the back-projection point. The back-projection process can be written as:(10)$${\tilde{q}}_{\mathrm{jik}}=\lambda_{ji}\cdot A\cdot (R_{\mathrm{ji}}\cdot P_{k}+T_{\mathrm{ji}})$$
where $\lambda_{ji}$ is a nonzero scale factor, A represents the intrinsic matrix of the camera, and $R_{\mathrm{ji}}=(I-2\cdot n_{\mathrm{ji}}\cdot {n_{\mathrm{ji}}}^{T})\cdot R_{s2cj}$, $T_{\mathrm{ji}}=(I-2\cdot n_{\mathrm{ji}}\cdot {n_{\mathrm{ji}}}^{T})\cdot T_{s2cj}+2\cdot d_{ji}\cdot n_{\mathrm{ji}}$.

Combined with the Equation (10), the reprojection error function of the back-projection can be expressed as:(11)$$Errpro=\sum_{j=1}^{W}\sum_{i=1}^{M}\sum_{k=1}^{N}{\left\|q_{\mathrm{jik}}-{\tilde{q}}_{\mathrm{jik}}(R_{s2cj},T_{s2cj},n_{\mathrm{ji}},d_{ji},P_{k})\right\|}^{2}$$

Let $P_{\mathrm{jk}}$ represent the k feature point of the j checkboard in the camera coordinate system.
(12)$$P_{\mathrm{jk}}=(R_{s2cj}\cdot P_{k}+T_{s2cj})$$

Since the reference plane can be regarded as a standard plane, the coplanar constraint of the W grids should be added. Let Perr represent the fitting effect evaluation value of plane fitting function: [fitresult,Perr]=createFit(dx, dy, dz). The input of the function is $P_{\mathrm{jk}}$. The smaller the Perr value is, the better the coplanar effect will perform. In addition, $R_{s2cj},j\in \{1,\ldots W\}$ are equal in theory. Let $R_{\mathrm{av}}$ represent the average rotation matrix [24]. The error Rerr between $R_{s2cj}$ and $R_{\mathrm{av}}$ can be written as:(13)$$\mathrm{Rerr}={\sum_{j=1}^{W}\left\|R_{s2cj}-R_{\mathrm{av}}\right\|}^{2}$$

The smaller the Rerr value is, the better the coplanar effect will perform. Likely, the five plane mirror positions with zero tilt angle on the optical platform also have coplanar characteristics. Therefore, the corresponding normal vectors $n_{j1}$ are theoretically equal. The average normal vector $n_{\mathrm{av}}$ can also be calculated.
(14)$$Nerr=\sum_{j=1}^{W}{\left\|n_{j1}-n_{\mathrm{av}}\right\|}^{2}$$

In the ideal condition, Perr=0, Rerr=0, Nerr=0. Therefore, the cost function can be regarded as two major components: the reprojection error term Errpro and the coplanar constraint term (Perr, Rerr, Nerr). We can establish the cost function in the case of equality constraints:(15)$$\left\{ \begin{array}{l} F=\min\sum_{j=1}^{W}\sum_{i=1}^{M}\sum_{k=1}^{N}{\left\|q_{\mathrm{jik}}-\tilde{q}(R_{s2cj},T_{s2cj},n_{\mathrm{ji}},d_{ji},P_{k})\right\|}^{2}+Errcop\\ Errcop=Perr+\mathrm{Rerr}+Nerr \end{array}\right.$$
where $R_{s2cj}$, $T_{s2cj}$, $n_{\mathrm{ji}}$ and $d_{ji}$ are parameters to be optimized. The calculation of the specific LM algorithm can be realized by the tool function lsqnonlin() in Matlab.

### 2.5. Three-Dimensional Measurement Principle of a Single Camera

In the monocular measurement system, we observe the images of the grid pattern, reflected in the unknown surface when the pose of the camera is known, and establish the reflection correspondence between the three-dimensional reference points and the two-dimensional image points. The depth of the reflection points on the surface is parameterized, and the surface shape is fitted by a polynomial. Therefore, the measurement of the surface shape is converted into an optimization problem: minimizing the error between the reference points and the corresponding points through the surface back projection [12]. The principle of the measurement system is shown in Figure 3. O is the origin of the camera coordinate frame, m is a feature point on the reference plane, p is a reflection point of the surface, and v is a projection point on the normalized image plane. p and v are called reflection correspondences. l is the reflected ray at p, and i is the incident ray. $R_{s2c}$ and $T_{s2c}$ are the rotation matrix and translation vector from reference plane coordinate frame to camera coordinate frame. Obviously, v is on the incident ray i. The relationship between p and v is given by
(16)p=s⋅v

s is the depth of the corresponding reflected point p. Correspondingly, the normal n to the surface at p can be written as:(17)$$n={\left(\frac{\partial p}{\partial x},\frac{\partial p}{\partial y},\frac{\partial p}{\partial z}\right)}^{T}$$

Suppose the coordinates of the normalized image points $\left\{v_{1},v_{1},\ldots ,v_{m}\right\}$ and points on the reference plane $\left\{m_{1},m_{2},\ldots ,m_{m}\right\}$ are known. The principle of back projection is shown in Figure 4. The three-dimensional reflection point on the mirror corresponds to the normalized image plane coordinates ${\left(x_{i},y_{i}\right)}^{T}$ that can be expressed as $p_{i}=s_{i}{\left(x_{i},y_{i},1\right)}^{T}$. The unit vector of the incident ray is $i_{i}={\left(x_{i},y_{i},1\right)}^{T}/\left\|{\left(x_{i},y_{i},1\right)}^{T}\right\|$, the unit vector of the reflected ray is $l_{i}=i_{i}-2\cdot \langle {\tilde{n}}_{i},i_{i}\rangle \cdot {\tilde{n}}_{i}$, and ${\tilde{n}}_{i}=n_{i}/\left\|n_{i}\right\|$. Let $R_{s2c}=\left(r_{1}\;r_{2}\;r_{3}\right)$, $r_{3}$ represents the coordinates of the unit vector in the *Z*-axis direction of the reference plane coordinate frame in the camera coordinate frame. $T_{s2c}$ indicates the coordinates of the origin of the reference plane coordinate frame in the camera coordinate frame. The reference plane can be represented by the vector $q={\left({r_{3}}^{T},-{r_{3}}^{T}\cdot T_{s2c}\right)}^{T}$, such that $\langle q,{\left({{\hat{m}}_{i}}^{T},1\right)}^{T}\rangle =0$ for any point on the reference plane. Back-projection can be achieved by computing the point $\hat{m}$, the intersection of the reflected ray with the reference plane.
(18)$${\hat{m}}_{i}=p_{i}-(\langle r_{3},p_{i}\rangle -{r_{3}}^{T}\cdot T_{s2c})/\langle r_{3},l_{i}\rangle \cdot l_{i}$$

In Equation (18), ${\hat{m}}_{i}$ is a function of depth s. We can build an optimization model to minimize the error between the back projection point and the real point on the reference plane. That means solving a nonlinear least-squares problem to estimate the depth of the mirror.
(19)$$\underset{s}{\min}{\sum_{i=1}^{m}\left\|{\hat{m}}_{i}(s)-m_{i}\right\|}^{2}$$

For minimizing problems in (19), we can also iteratively calculate s with the LM algorithm. The initial surface can be regarded as a plane.

## 3. Experimental Verification

### 3.1. Calibration Experiment

To verify the accuracy and universality of the calibration method proposed in this paper, a monocular vision system measurement experiment was designed (Figure 5). The whole measurement system consists of an optical platform, standard plane mirror, LCD monitor, and large FOV camera. The focal length of the camera is 8 mm; the image resolution is 1280 pixel × 1024 pixel, and the pixel size is 4 μm; the FOV of the camera is 820 mm × 670 mm, which is much bigger than grid image. When the measurement distance is about 1000 mm, the field of view of the camera is 820 mm × 670 mm. The LCD is 19 inches in size and has a pixel size of 0.2451 mm. In order to approach a large field of view measurement scene, we use a 90 × 120 mm checkerboard image as a calibration target, which is much smaller than the camera’s field of view range.

The LCD faces the standard plane mirror on the optical platform. The grid image on the LCD is captured by the camera through the plane mirror. In the experiment, the grid image is moved on the LCD. Each grid position corresponds to three positions of a plane mirror, which are the position STZ on the optical platform, the position STX around the *X*-axis, and the position STY around the *Y*-axis. In this way, it not only ensures that the three positions of the plane mirror intersect with each other to satisfy the orthogonality constraint, but also ensures that there is an obvious height difference to satisfy the conditions of Zhang’s calibration method.

Figure 6 is a set of mirrored images of the grid taken by the camera for calibration. The grid image was moved five times, and the five positions of the grid basically filled the whole LCD screen to cover the whole FOV of the camera. In the five pose conversion parameters from the reference coordinate system to the camera coordinate system, the rotation matrices are equal, and the translation vectors change with the motion of the grid in theory. In the same way, the plane mirrors at the STZ position corresponding to the five grid images are also coplanar, so the corresponding mirror normal vectors are equal. This is the coplanar constraint described in Section 2.4.

Figure 7 describes the mirrored grid positions and the real grid positions with and without coplanar constraints. The mirrored grid positions corresponding to the STZ positions of the five plane mirrors are coplanar. Fitting the plane of five mirrored grid positions, the average distance error RMSE is 0.14 mm, which is consistent with Figure 7a,c. However, the coplanarity of the five grid positions restored is obviously different.

As is shown in Figure 7c,d, the positions of each chessboard are not only poor in coplanarity, but they also have a large offset in the relative positions, which can not comply with the law of mirror reflection.

Figure 8a is the coplanarity of the five grids performs well with coplanar constraint, RMSE = 0.11 mm. However, the five grids without coplanar constraint have poor coplanarity, RMSE = 6.45 mm. Figure 8b is the reprojection error of the two methods after back projection. The average reproject error of the method proposed in this paper is 0.1641 pixels, and in paper [16], it is 0.1419 pixels. 

The two methods are similar in terms of calibration accuracy, and the reprojection error without coplanar constraint is smaller. However, for the reference plane, the calibration result of this method is locally optimal. With coplanar constraints, the reprojection optimization model can unify the positions of five checkerboards and optimize the calibration results as a whole. Therefore, the calibration method in this paper sacrifices part of the calibration accuracy to improve the reliability of the algorithm. This calibration result is more suitable for practical measurement.

### 3.2. Measurement of the Step Surface

After the calibration of the reference plane, we can carry out a three-dimensional measurement experiment according to Section 2.5. As is shown in Figure 9a,b, a standard plane mirror is placed on the optical platform, and the mirror feature point calculation is performed at the STZ position. Then place the standard gauge block between the optical table and the planar mirror, so the mirror position is 8.74 mm higher than before, and the mirror feature points are calculated at the higher mirror position. Fit the mirror surface with feature points by createFit(), and then use the point-to-plane distance formula to calculate the distance from each feature point to the fitting plane, and then take the average value. Compare it with the actual distance of 8.74 mm to indirectly verify the accuracy of the calibration method proposed in this paper. The mirror feature points of the first mirror position are shown in Figure 9c. The plane fitting model is as follows:
(20)$$f(x,y)=p_{00}+p_{10}\cdot x+p_{01}\cdot y$$

We can obtain the coefficients of the plane: $p_{00}=421.4000,\;p_{10}=-0.6167,\;p_{01}=0.0267$, and the RMSE = 0.02 mm. To have an intuitive display effect, the first and second mirror positions are shown together in Figure 9d. The average distance of the two mirror positions is 8.68 mm. The difference with the actual distance of 8.74 mm is 0.06 mm, and the relative error is 0.69%. 

### 3.3. Measurement of the Spherical Mirror

In addition, we also measure the spherical mirror surface. The principle of the experiment is the same as that of the mirror. Firstly, measure five sets of spherical characteristic points, with 108 points in each group as measurement data. Then, place the spherical mirror on a coordinate measuring machine (model: MC850) with the highest resolution of 1 um for sampling.

The number of detection points is 202, which is used as reference data. Since the coordinate system of the coordinate measuring machine is not unified with the camera coordinate system, it is necessary to use Cloud-Compare software to unify the measurement data and reference data with the method of iterative closest point (ICP). The ICP registration of the measured feature points and the reference feature points is shown in Figure 10b.

The spherical equation is fitted to the reference data through Cloud-Compare software. As shown in Figure 11a, the spherical equation is:
(21)$$z=-459.621+\sqrt{\left(475.617^{2}-{\left(x-0.506226\right)}^{2}-{\left(y-0.264729\right)}^{2}\right)}$$

Additionally, RMSE = 0.01 mm. The fitting error distribution is shown in Figure 11b. We can obtain the spherical mirror radius from the Equation (21) (Table 1). 

In the experiment, we use a cubic polynomial to initialize the spherical mirror surface because we treat the mirror surface as unknown. Supposing we directly use the spherical equation to iteratively optimize the mirror surface, the measurement accuracy will perform better.

## 4. Conclusions

This paper proposes a calibration method based on coplanar constraints for a camera with a large FOV. The whole experiment process is divided into two parts. The first is the calibration of a large FOV camera and the reference plane. By adjusting the tilt angle of the planar mirror and moving the grid image on the LCD monitor, the camera acquires multiple sets of calibration images and then obtains the optimal solution of the external parameters between the camera and the LCD monitor with the coplanar constraint. The other is shiny surface reconstruction. When the pose of the reference plane is known, we can establish the dense reflection correspondence between normalized image plane two-dimensional feature points, reference plane three-dimensional feature points, and bright surface reflection points, and we can iteratively calculate the reflection point depth information. In terms of calibration accuracy, the calibration accuracy of the method proposed in this paper is similar to that of [16]. At the same time, in the step surface and spherical surface measurement experiments, the results also indirectly prove the accuracy of the proposed method. The universality of the method has important research significance for further application to the multi-camera measurement system in the future.

## Figures and Tables

**Figure 1 sensors-23-03464-f001:**
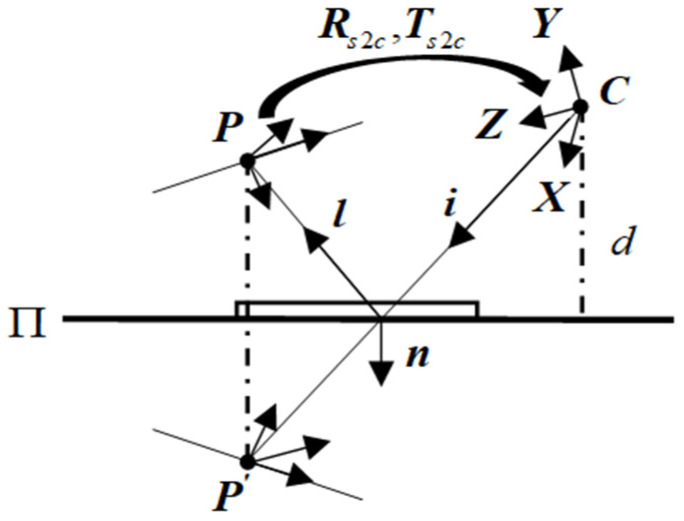
Calibration principle for reference plane. The camera C observes a point P on the reference plane via the plane mirror Π. We denote by i the incident ray and by l the reflected ray, $R_{s2c}$ and $T_{s2c}$ denote the pose parameters between the reference plane and the camera, n denotes the normal of the mirror, and d is the distance between C and Π.

**Figure 2 sensors-23-03464-f002:**
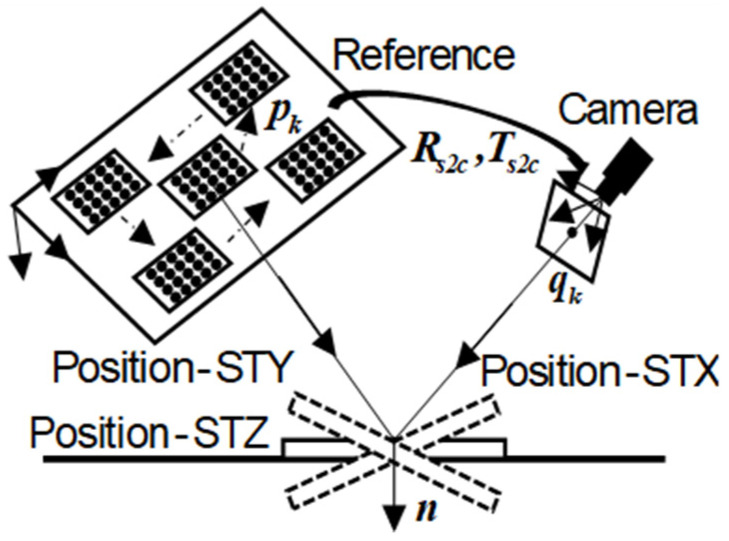
Structure of measurement system. The calibration grid moves on the reference plane for M times. A feature point $P_{k}$ of the grid is reflected in the image point $q_{k}$ via the mirror at position STX, STY, STZ. We can obtain three calibration images at each grid location. Then, W×3 calibration images can calculate the intrinsic matrix A, as well as the pose parameters $R_{\mathrm{ji}}$ and $T_{\mathrm{ji}}$. Finally, $R_{s2c}$, $T_{s2c}$, n, and d can be calculated by Equations (8) and (9).

**Figure 3 sensors-23-03464-f003:**
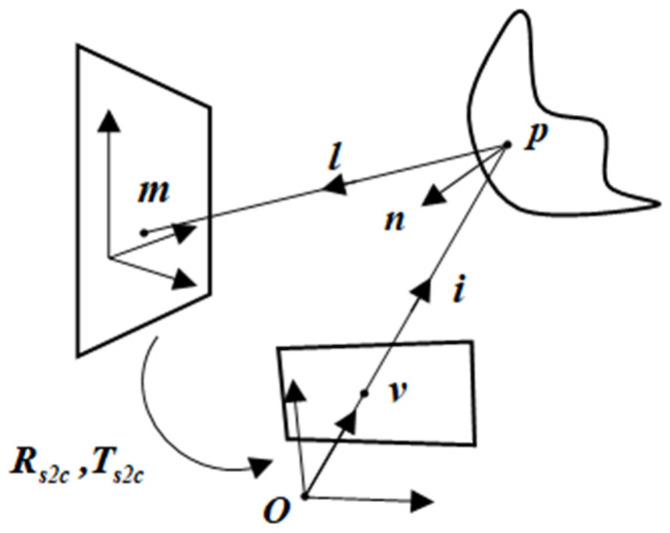
Principle of mirror surface measurement. A pinhole camera centered at O is observing a mirror surface point p that reflects a reference point m to an image point v. We refer to m and v as reflection correspondences. The reflected ray l is determined by m and p. We denote by i the incident ray for image point v and by n the normal at p. $R_{s2c}$ and $T_{s2c}$ denote the pose parameters between the reference plane and the camera.

**Figure 4 sensors-23-03464-f004:**
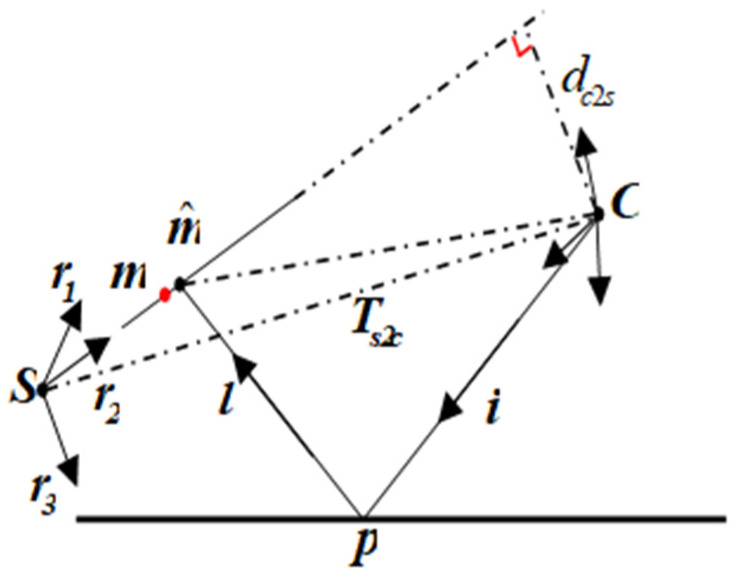
Principle of back projection. The rotation matrix $R_{s2c}$ can be written as $\left(r_{1}\;r_{2}\;r_{3}\right)$. $r_{3}$ denotes the unit vector in the *Z*-axis of the reference plane. $T_{s2c}$ denotes the distance between S and C. The reflected ray l intersects the reference plane at the point $\hat{m}$. The point $\hat{m}$ satisfies $-{r_{3}}^{T}\cdot \hat{m}=d_{s2c}$. We denote by $d_{c2s}$ the distance between C and the reference plane.

**Figure 5 sensors-23-03464-f005:**
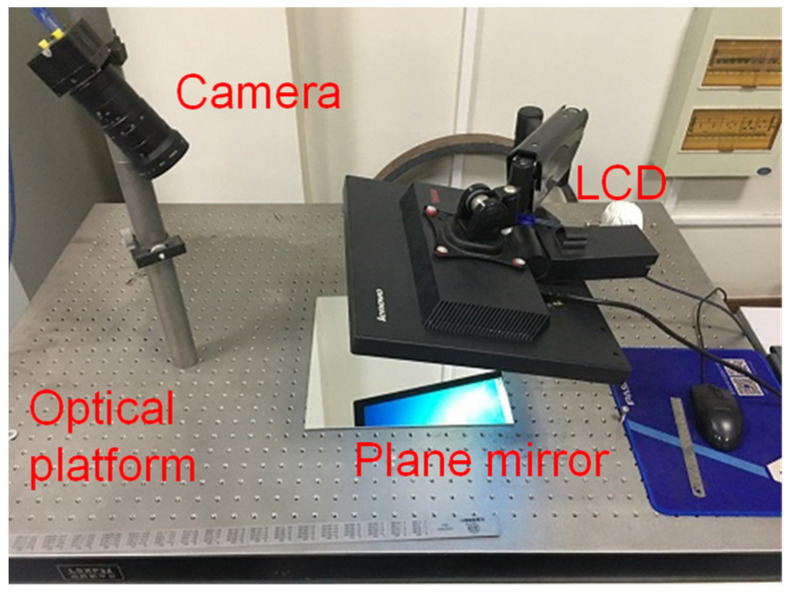
Experiment setup. The whole measurement system consists of an optical platform, standard plane mirror, LCD monitor, and large FOV camera.

**Figure 6 sensors-23-03464-f006:**
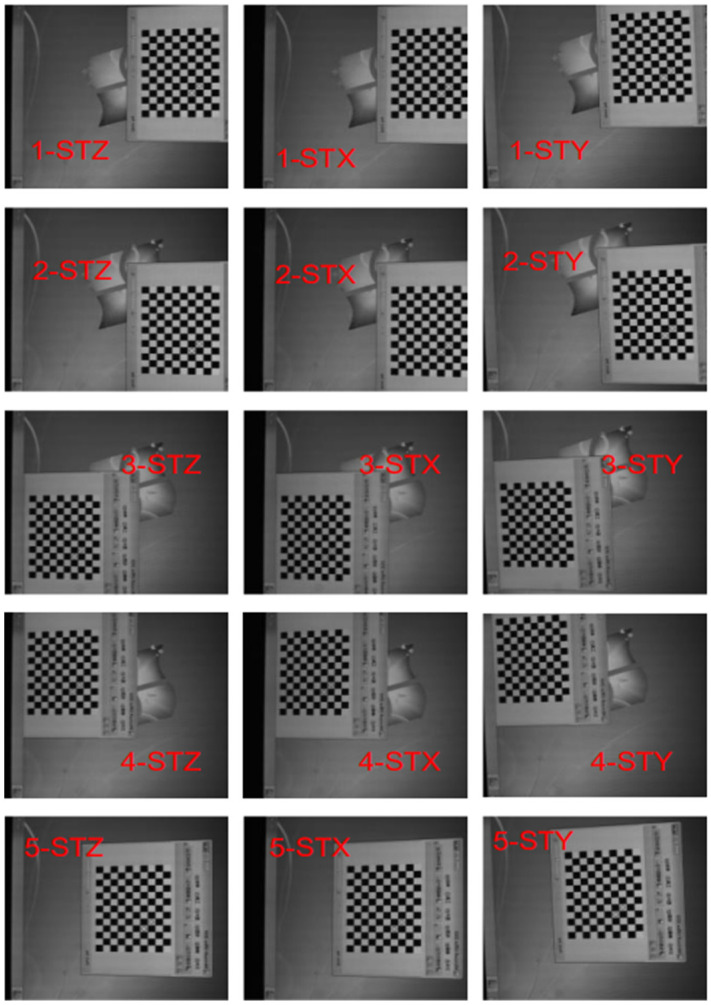
Checkerboard mirror image taken by the camera for calibration. The calibration grid moves on the reference plane five times. We change the position of the plane mirror to position STX, STY, and STZ for each grid. Then, we can obtain 5×3=15 calibration images.

**Figure 7 sensors-23-03464-f007:**
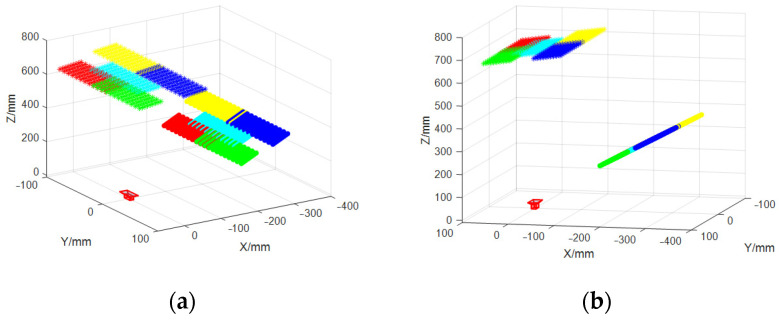
Calibration results with and without coplanar constraints. • points denote the real feature points on the reference plane. ∗ points denote the mirrored feature points. (**a**) Calibration results with coplanar constraints in view 1. (**b**) Calibration results with coplanar constraints in view 2. (**c**) Calibration results without coplanar constraints in view 1. (**d**) Calibration results without coplanar constraints in view 2. Yellow: Grid location 1-STZ. Blue: Grid location 2-STZ. Green: Grid location 3-STZ. Red: Grid location 4-STZ. Cyan: Grid location 5-STZ.

**Figure 8 sensors-23-03464-f008:**
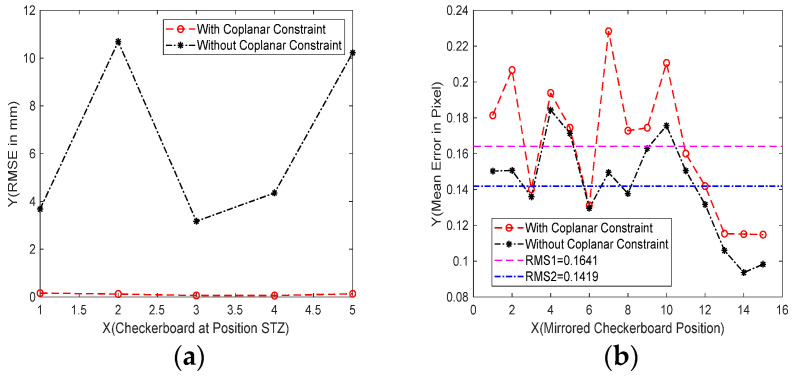
Comparison of error with and without constraints. (**a**) Coplanarity error of the two methods. With constraints: RMSE = 0.11 mm. Without constraints: RMSE = 6.45 mm. (**b**) Reprojection error of the two methods. With constraints: RMS = 0.1641 pixel. Without constraints: RMS = 0.1419 pixel.

**Figure 9 sensors-23-03464-f009:**
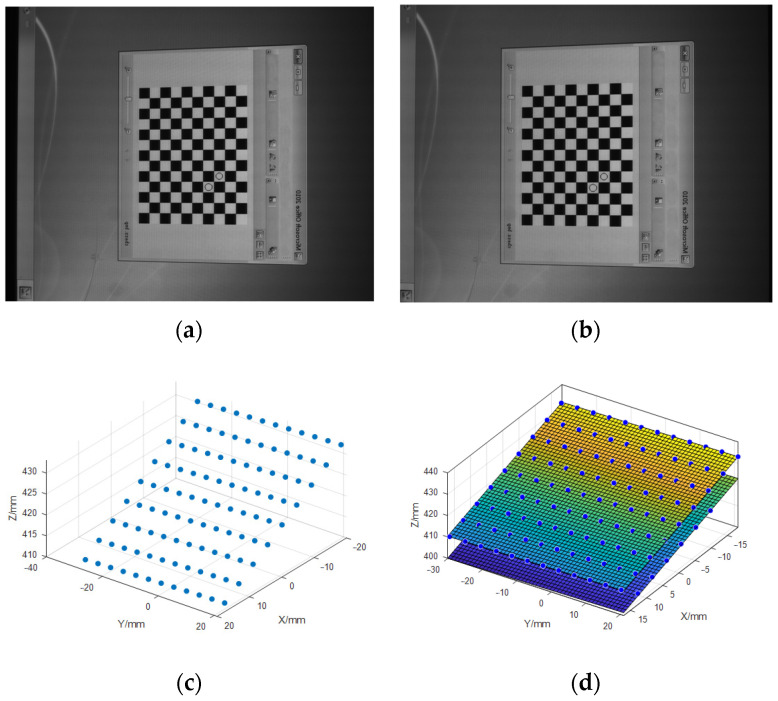
Restoration of plane mirror feature points at position 5. (**a**) Mirrored checkerboard image before placing the standard gauge block. (**b**) Mirrored checkerboard image after placing the standard gauge block. (**c**) Mirror feature points at position STZ. (**d**) Plane fitting of two mirror feature points. The distance between the two planes is 8.68 mm.

**Figure 10 sensors-23-03464-f010:**
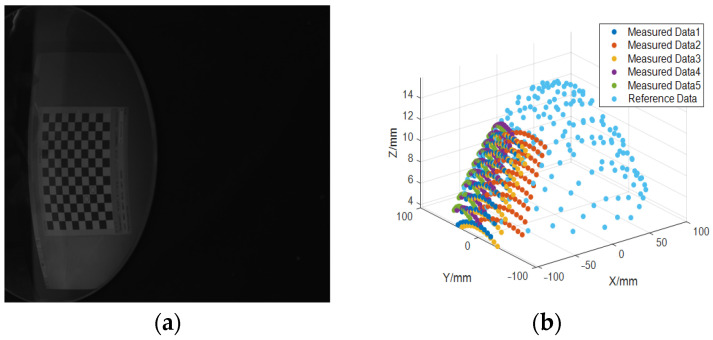
Restoration of spherical mirror feature points. (**a**) Mirrored checkerboard image of the spherical mirror at position 5. (**b**) Display of feature points and reference data.

**Figure 11 sensors-23-03464-f011:**
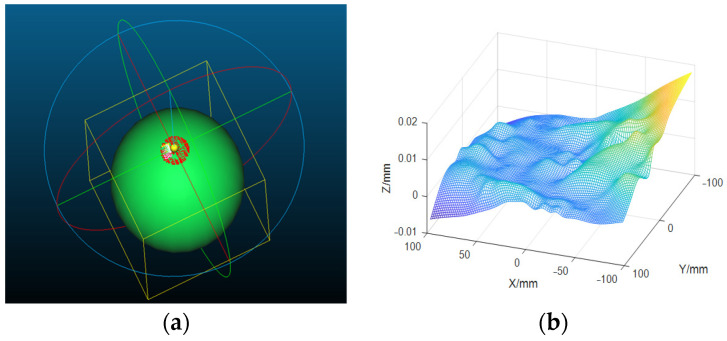
Spherical equation fitting of reference points. (**a**) Spherical equation fitting. The fitting result: radius of the spherical mirror is 475.62 mm. RMSE = 0.01 mm, the manufacturing error can be ignored. (**b**) Surface fitting error.

**Table 1 sensors-23-03464-t001:** The fitting results of measurement data and the measurement error results compared with the reference data.

Point-Data	Radius (mm)	RMSE (mm)	Error (%)
Data 1	477.94	0.02	0.49
Data 2	478.32	0.02	0.57
Data 3	472.48	0.02	0.66
Data 4	478.54	0.01	0.61
Data 5	472.37	0.02	0.68

## Data Availability

Data will be made available on request.

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
