# Peer review of "Accurate Calibration of a Large Field of View Camera with Coplanar Constraint for Large-Scale Specular Three-Dimensional Profile Measurement"

_sensors, 2023, doi:10.3390/s23073464_

Round 1

Reviewer 1 Report

The paper describes a camera calibration method using a mirror and a set of coplanar targets. The actual contribution of the paper resides in the complanar penalty added during the LM refinement step of extrinsic parameters.

I have some concerns that need to be addressed:

- As a general motivation for the work, the impracticability of a standard checkerboard-based calibration technique is argued. Since the proposed method requires a non-trivial setup, (mirror + monitor + stand), is it really more practical?

- Also, the authors argue that they calibrate a large FOV camera, but from the specs provided (And also from the images), it looks quite narrow (about 50 deg?). If this is the case, I would remove the case and reformulate the introduction to motivate this work better. The title needs to reflect the actual contribution, too.

- On the methodological part, it is not clear how you deal with the scale factor. Is it based on the monitor pixel size? Is it, in general, a good calibration target for high-precision measurements?

- Since the authors use a single monitor for calibration, other monitor-based calibration techniques need to be discussed. In particular, is it necessary to have multiple targets when dense patterns for monitor calibration exist? (see Adopting an Unconstrained Ray Model in Light-Field Cameras for 3D Shape Reconstruction, CVPR 2015 and Parameter-free Lens Distortion Calibration of Central Cameras, ICCV 2017). As an alternative, wouldn't occlusion-resistant markers such as Charuco (Charuco Board-Based Omnidirectional Camera Calibration Method) or RUNETag (An Accurate and Robust Artificial Marker Based on Cyclic Codes, TPAMI 2016) be an option?

- Other relevant work: coplanarity of calibration targets has also been investigated in other relevant works that need to be discussed, such as (Camera calibration from coplanar circles, ICPR 2014; Camera calibration from very few images based on soft constraint optimization, 2020; Camera Calibration with Two Arbitrary Coplanar Circles, ECCV 2004 )

- Eq 15 does not look quite right. Isn't the optimization problem to minimize Errcop? Why is it subtracted? Also, how are lambdas chosen? Sensitivity to these hyperparameters should be shown in the experimental part.

- how is the relative error computed in tab 1?

- row 100: you forgot to delete some text from the original template. (quite worrying)

Author Response

Dear Sir:

Thank you for your guidance on my article, which made me find many imperfections. Now, I will reply to your comments one by one:

1.The calibration method proposed in this paper is mainly applied to the measurement scene of highly reflective objects, that is, the calibration link in the phase-shift deflection method. In this scene, it is usually necessary to project the coded image from the display to the measured surface, and capture the modulated image reflected by the bright surface by the camera. Therefore, in the deflection method scene, the display and bracket are very common; In this scenario, because of the nature of the bright surface, the camera cannot directly observe the display screen target, so it is usually necessary to use a plane reflector to make the camera observe the display indirectly to complete the calibration.

2.As mentioned above, this paper is mainly aimed at the measurement scene of highly reflective objects, that is, the phase-shifting deflection method scene. In the current common deflection method scene, the camera's field of view is usually small. And the large field of view described in this paper is more of a relative concept, that is, to calibrate a camera with a larger field of view by using the coplanarity of multiple objects. Therefore, the methods and conclusions of this paper can be better transplanted to camera calibration with a larger field of view.

3.In the experiment, the penalty function factor is generally 1 by default. To avoid confusion among readers, it is deleted.

  1. An important reason for using the monitor for calibration in this paper is that the monitor is not only the marker used for calibration, but also the carrier of fringe projection used for phase-shift deflection measurement. Therefore, other special anti-occlusion calibrators are not considered in this paper. At the same time, if dense targets are directly used instead of multiple targets, I think it will reduce the robustness of the calibration results, and the large area of fixed dense targets will also limit the placement relationship between the camera and the monitor, which will undoubtedly affect the flexibility of the phase-shift deflection system measurement.
  2. According to your suggestion, the article adds new references and describes them in the introduction to illustrate the feasibility of the study.
  3. The relative error in Table 1 refers to the relative error between the radius of the sphere fitted by the point cloud and the radius of the real spherical mirror

7.The calculation method of relative error is added in the experiment.

Thank you again for your valuable comments.

Your sincerely

Wang Zhizhuo

Reviewer 2 Report

1) The introduction section must highlight the drawbacks of other works. Now, it looks like a summary

2) What is the need of so many mathematical equations? Whether all these are needed? Justify it

3) How do you initialize the variables used in the equations?

4) The methodology part is very weak. It doesn't talk much about the proposed method. More theoretical information must be added.

5) What is the scenario of current calibaration? How do you say that your proposed method is better than the current method?

6) What do you infer from Figure 7?

7) What is the ideal value of error available with the conventional methods? Do you think your result is significant?

8) If you change the mirror conditions, will the results change? How far your assumptions hold true in real-time situations?

Author Response

Dear Sir:

Thank you for your guidance on my article, which made me find many imperfections.

  1. According to your suggestion, the introduction is revised, new references are added and their deficiencies are introduced
  2. I think the detailed mathematical formula and derivation will help readers and peers understand it
  3. This paper tries to simplify the description of the existing and other literature methods, that is, how to get the rough estimation of camera parameters and the reconstruction process for verification; The coplanar constraint, the main work of this paper, is described in detail
  4. The current calibration situation is described in the introduction, and the work of this paper has made some improvements to its shortcomings.
  5. The method proposed in this paper is relatively robust when the mirror condition changes. In the experiment, the plane mirror angle is changed many times, but the error does not fluctuate significantly.

Thank you again for your valuable comments.

Your sincerely

Wang Zhizhuo

Round 2

Reviewer 1 Report

Dear authors, 

Thanks for answering my doubts. Nevertheless, I think it would be beneficial for the paper if the answers provided were also reflected in major changes in the text.  In particular, the authors should include the following points in the introduction:

- the motivation why monitor-based calibration is a standard in the measurement of highly reflective objects.

- a motivation and some references on why a 50deg FOV has to be considered a large field of view in this specific application.

- a discussion of other possible calibration methods (the one mentioned in the previous review) and why they are not applicable (the simplicity of the setup could be a motivation, but still, they need to be discussed). Consider that dense calibration targets, such as the CVPR 2015 and 2017, are highly robust to partiality (indeed, each pixel is independent), but require multiple frames. RUNETag and ChAruco are also robust to partiality but present fewer points for calibration.

Also, I did not receive an answer about the scale factor calibration. 

Should the authors modify the paper according to their answers to the previous reviews and my comments, I would be happy to recommend the paper for publication.

Author Response

Dear Editors and Reviewers:

Thank you for your letter and for the reviewer’s comments concerning our manuscript entitled “Accurate calibration of a large field of view camera with co-planar constraint for large-scale specular 3D profile measurement”. Those comments are all valuable and very helpful for revising and improving our paper. We have studied comments carefully and have made correction which we hope meet with approval. Revised portion are marked in red in the paper. The main corrections in the paper and the responds to the reviewer’s comments are as flowing:

  1. The motivation why monitor-based calibration is a standard in the measurement of highly reflective objects:

Response: According to your suggestion, about “the motivation why monitor-based calibration is a standard in the measurement of highly reflective objects” has been described in the citation[66-76].

  1. A motivation and some references on why a 50deg FOV has to be considered a large field of view in this specific application:

Response: The coplanar constraints proposed in this paper are universal and can be well applied to camera systems with larger field of view angles. The camera system used in the experiment has a 50 ° FOV, which is larger than the calibration grid used, which also reflects the universality of the conclusions of this article. We have added relevant introductions to the experiment [246-251]and the end of the article.

3.A discussion of other possible calibration methods (the one mentioned in the previous review) and why they are not applicable (the simplicity of the setup could be a motivation, but still, they need to be discussed). Consider that dense calibration targets, such as the CVPR 2015 and 2017, are highly robust to partiality (indeed, each pixel is independent), but require multiple frames. RUNETag and ChAruco are also robust to partiality but present fewer points for calibration:

Response: The introduction adds a discussion of the relevant literature you provided [66-76]. At the same time, after our discussion, we believe that ,in the manufacturing process of display screens, even more expensive industrial displays often have a certain curvature (this is not an ideal plane) due to production process constraints. The larger the size, the more pronounced the curvature. This is a disadvantage of LCD monitors compared to professional ceramic calibration plates. This is understandable in daily use, but in deflection method scenarios, when using a large size display for dense calibration, the curvature of the display screen itself will mask the accuracy improvement brought about by dense calibration, but for relatively sparse small targets, the impact of the curvature of the display screen itself will be reduced. Therefore, we use small targets at different locations to cover the camera's field of view, thereby completing the calibration of larger field cameras in an offset field of view.

4.Also, I did not receive an answer about the scale factor calibration:

Response: Regarding the setting of penalty function factors, since the cost function is divided into two parts, namely, re projection and coplanar constraints, adding a penalty function factor before these two terms allows the cost function to iterate in the direction of more biased re projection or coplanar constraints based on the ratio of penalty function factors during the convergence process. However, during the experiment, the ratio of the two penalty function factors has a small impact on the results, so the numerical value of both terms in the experiment section of this article is 1. Sorry for causing trouble to your reading, so this article has deleted the content about penalty function factors.

Finally, thank you for your valuable comments on this article. We are deeply impressed by the agility and depth of your research, and your suggestions have played an important role in the revision of this article. Wish you a happy life!

Your sincerely

Zhizhuo Wang

Reviewer 2 Report

It can be accepted. It is improved

Author Response

Dear Editors and Reviewers:

Thank you for your letter and for the reviewer’s comments concerning our manuscript entitled “Accurate calibration of a large field of view camera with co-planar constraint for large-scale specular 3D profile measurement”. Those comments are all valuable and very helpful for revising and improving our paper. 

Your sincerely

Zhizhuo Wang

Round 3

Reviewer 1 Report

Dear Authors, 

thanks for taking into account my suggestions. I have just a couple of comments on your answers to my previous review:

- I agree that the proposed method should adapt to a large FOC camera s well, but since the paper put a lot of focus on a large field of view (also in the title), I would have expected to see this reflected in the experimental part. Nevertheless, if other reviewers and the editor are fine with this, I won't insist further.

- with scale factor, I do not mean the weighting of the terms in the loss, but the "scaling factor" from the camera world to the real world that needs to be calibrated. Since the checkerboard you use is in pixels, you should know the pixel size of the monitor to use the checkerboard for this purpose, which is not always a piece of information trivial to retrieve. As an alternative, you can use a calibration reference object with known dimensions.

Author Response

Dear Editors and Reviewers:

Thank you for your letter and for the reviewer’s comments concerning our manuscript entitled “Accurate calibration of a large field of view camera with co-planar constraint for large-scale specular 3D profile measurement”. Those comments are all valuable and very helpful for revising and improving our paper. We have studied comments carefully and have made correction which we hope meet with approval. Revised portion are marked in red in the paper. The main corrections in the paper and the responds to the reviewer’s comments are as flowing:

Responds to the reviewer’s comments:

1.- I agree that the proposed method should adapt to a large FOC camera s well, but since the paper put a lot of focus on a large field of view (also in the title), I would have expected to see this reflected in the experimental part. Nevertheless, if other reviewers and the editor are fine with this, I won't insist further.

Response:

In the experimental section of the article, we have added some new statements to prove the rationality of the "large field of view" described in the article, mainly including increasing the target size data and emphasizing the size comparison relationship with the camera's field of view. After discussion, another reviewer and editor were satisfied with this.

2.- with scale factor, I do not mean the weighting of the terms in the loss, but the "scaling factor" from the camera world to the real world that needs to be calibrated. Since the checkerboard you use is in pixels, you should know the pixel size of the monitor to use the checkerboard for this purpose, which is not always a piece of information trivial to retrieve. As an alternative, you can use a calibration reference object with known dimensions.

Response:

1)Regarding the scale factor, it is the reciprocal of the value of the target point in the Z direction of the camera coordinate system. To reduce unnecessary narration, we simplify it to a scale factor. This representation is very common.

2)In the experimental section of the article, we have given the pixel sizes of the display and camera sensors, and added the size of the target. As the size of the target is given, we believe that the field of view of the camera relative to the target is already large enough, which also makes our discussion of "large field of view" reasonable.

Finally, thank you for your valuable comments on this article. We are deeply impressed by the agility and depth of your research, and your suggestions have played an important role in the revision of this article. Wish you a happy life!

Your sincerely

Zhizhuo Wang